



# Comparison of Flood Inundation Modeling Frameworks within a Small Coastal Watershed during a Compound Flood Event

Joseph L. Gutenson[1,2], Ahmad A. Tavakoly[1,3], Mohammad S. Islam[4], Oliver E. J. Wing[5], William P. Lehman[6], Chase O. Hamilton[1], Mark D. Wahl[1], T. Christopher Massey[1]

[1]U. S. Army Engineer Research and Development Center, Coastal and Hydraulics Laboratory, Vicksburg, MS, 39180, United States
[2]Department of Civil, Construction, and Environmental Engineering, The University of Alabama, Tuscaloosa, AL, 35487, United States
[3]Earth System Science Interdisciplinary Center, University of Maryland, College Park, MD 20740, United States
[4]U. S. Army Corps of Engineers, Galveston District, Galveston, TX, 77550, United States
[5]Fathom, Bristol, United Kingdom
[6]U. S. Army Corps of Engineers, Hydrologic Engineering Center, Davis, CA, 95616, United States

*Correspondence to*: Joseph L. Gutenson (jlgutenson@gmail.com)

**Abstract.** The flooding brought about by compound coastal flooding can be devastating. Before, during, and immediately following these events, flood inundation maps, or Events Maps, can provide essential information to emergency management. However, there are a number of frameworks capable of estimating Event Maps during flood events. In this article, we evaluate three such Event Map frameworks in the context of Hurricane Harvey. Our analysis reveals that each of the three frameworks provide different inundation maps that differ in their level of accuracy. Each of the three Event Maps also produce different exposure and consequence estimates because of their physical differences. This investigation highlights the need for a centralized means of vetting and adjudicating multiple Event Maps during compound flood events empowered by the ability to distribute Event Maps as geographic information system (GIS) services and coalesce Event Maps into a common operating picture. Furthermore, we provide evidence that the ability to produce multi-model estimates of Events Maps to create probabilistic Event Maps may provide a better product than the use of a lone Event Map.

**Short summary**

Emergency managers use event-based flood inundation maps, or Event Maps, to plan and coordinate flood fights. We perform a case study test of three different flood mapping frameworks to see if the Event Map differences lead to substantial differences in the location and magnitude of flood exposure and consequences. We find that the Event Maps are much different physically and that the physical differences do produce differences in the location and magnitude of exposure and consequences.



# 1 Introduction

Each year, tropical storms devastate portions of the coastal United States. From 1980-2020, tropical storms accounted for $945.9 billion in damages with an average of $21.5 billion in damages per event (Fast Facts: Hurricane Costs, 2021). Tropical storms bring strong winds and heavy rainfall that are the primary drivers of compound flooding. Strong winds and high tide

create storm surge, pushing coastal waters inland and inundating land that is typically dry. Inland, heavy rainfall leads to direct runoff and saturation excess runoff from the land surface into inland waterbodies. The combination of inland runoff and storm surge creates compound coastal flooding. Recent studies highlight how the combination of inland drainage and coastal surge are important in properly estimating compound floods (Gori et al., 2020; Loveland et al., 2021).

In order to inform emergency managers and the public at-large, agencies such as the National Oceanic and Atmospheric

Administration's (NOAA's) National Weather Service (NWS), the U. S. Army Corps of Engineers (USACE), the Federal Emergency Management Agency (FEMA), and the U. S. Geological Survey (USGS) produce estimates of flood inundation for inland, coastal, and compound flood events. The Integrated Water Resources Science and Services (IWRSS) refers to such flood inundation maps as Event Maps (IWRSS, 2013; 2015). Event Maps are help emergency managers communicate situational awareness, devise response plans, and inform decision makers (NWS, 2012; IWRSS, 2013; Maidment, 2017;

Longenecker et al., 2020). However, data availability to create Event Maps can vary dramatically across the world and can originate from a number of sources. The disparate origins of multiple Event Maps for an event can create unnecessary confusion and conflicted decision making for decision makers.

A number of frameworks and methodologies exist to create accurate Event Maps. For inland fluvial flooding, NOAA's National Water Center (NWC) co-developed and implemented the height above nearest drainage (HAND) inundation model

that uses the Manning's equation to precompute inundation libraries to couple with hydrologic forecasts from the National Water Model (NWM) (Liu et al., 2018; Zheng et al., 2018; Viterbo et al., 2020). The HAND methodology requires a minimal amount of input data that are available over large geographic scales. Alternatively, USACE developed the AutoRoute model that functions in a similar manner to the NWC's HAND implementation, requiring minimal inputs, making it capable of producing flood inundation maps over continental-scale geographic extents (Follum 2013; Follum et al., 2016; 2020; Tavakoly

et al., 2021). HAND and/or AutoRoute perform well as first order approximations of fluvial flooding (Afshari et al., 2018; Johnson et al., 2020). However, these low complexity models do possess less skill when compared to higher fidelity hydraulic models (Hocini et al., 2021). One of the more notable limitations of steady-state inland models such as HAND and AutoRoute is their limitations in coastal watersheds. HAND and AutoRoute are fluvial-only flood models and their Event Maps do not inherently contain the pluvial or coastal components of flooding. Further, coastal watersheds tend to have minimal topographic

relief where one-dimensional (1D) models, such as HAND and AutoRoute, traditionally struggle to produce accurate flood inundation maps. Low topographic relief tends to create backwater effects that AutoRoute cannot physically account for (Follum et al., 2016; 2020). Further, where topographic relief is low HAND can be sensitive to errors in the underlying terrain



(Johnson et al., 2020). Thus, steady-state hydraulic models, such as HAND and AutoRoute, tend to have limited effectiveness in providing Event Maps during compound coastal floods in coastal watersheds.

For coastal flooding, NOAA's National Hurricane Center (NHC) produces Event Maps that estimate coastal flooding from storm surge using the the Sea, Lake, and Overland Surges from Hurricanes (SLOSH) model (Jelesnianski et al., 1984; Experimental Potential Storm Surge Flooding Map, 2022). The Coastal Emergency Risk Assessment (CERA) team creates coastal flooding only Event Maps using the Advanced Circulation (ADCIRC) model (Luettich et al., 1992; About: See the Storm Surge in Real-Time, 2022). However, these modeling frameworks do not currently include a coupling with inland

runoff.

In response to the limitations of existing fluvial and coastal flood mapping frameworks, Wing et al. (2019) use the Fathom-US large-scale hydraulic modeling framework (Wing et al., 2017) to perform Event Map estimation for Hurricane Harvey. The Wing et al. (2017) framework can account for coastal, fluvial, and pluvial flooding. Wing et al. (2019) compare the Fathom-US flood inundation results to the NWC HAND methodology. Wing et al. (2019) find that the Fathom-US framework

is more accurate than the NWC HAND methodology for the Hurricane Harvey simulations due to better representation of the complex physics that occur during compound coastal floods.

Beyond the large-scale modeling frameworks such as the NWC HAND or Fathom-US, there are local-scale compound flood models in data rich environments that can have higher spatiotemporal resolution and are capable of producing Event Maps that combine coastal, fluvial, and pluvial flooding. For example, the USACE Models, Mapping, and Consequences (MMC)

Production Center will work with local USACE districts and divisions to create and distribute Event Maps during flood events using existing Corps Water Management System (CWMS) model frameworks or develop new model frameworks on-the-fly (Winders et al., 2018). The simulation times of these frameworks can be a hindrance in their ability to produce a timely Event Map. However, these models can provide a benchmark for what is achievable with increased model fidelity and resolution. Further, we may be able to more effectively utilize these high fidelity simulations for Event Maps through surrogate modeling

techniques (Bass and Bedient, 2018; Zahura et al., 2020; Contreras et al., 2020; Kyprioti et al., 2021), similar to how the NWC-HAND and Fathom-US utilize a precomputed riverine hydraulics in those implementations (Zheng et al., 2018; Wing et al., 2019).

This paper seeks to investigate if different modeling frameworks produce substantially different Event Maps during compound coastal flood events. We evaluate and quantify the differences by using a Hurricane Harvey case study where a recently

developed local scale framework exists and compare this to the AutoRoute and Fathom-US frameworks. Hurricane Harvey is a now infamous compound flood event brought about by a combination of wet antecedent conditions, heavy inland rainfall, and sustained high water levels at the coast (Valle-Levinson et al., 2020). Our comparison of the three frameworks centers on the physical differences in each Event Map and if those differences lead to differences in estimated exposure and consequences. To our knowledge, this is the first evaluation of Event Maps produced with different flood map frameworks that seeks to

evaluate differences in the Event Maps by examining both the physical differences in the Event Maps and the estimated exposure and consequences from those Event Maps.



## 2 Methodology

To perform our comparison, we applied a recently developed unsteady hydrologic and hydraulic modeling in the Clear Creek watershed, south of Houston, Texas. Figure 1 demonstrates the location of the Clear Creek watershed that covers an area is roughly 698.91 km$^2$. The region has a history of repeated flooding, including flooding during Hurricane Harvey, and is subject to rapid development and urbanization (Brody et al., 2018).

**Figure 1: The Clear Creek watershed test domain for this study spans portions of four counties in Texas. Sources of the background imagery include Esri, TomTom, U. S. Department of Commerce, and Census Bureau.**



## 2.1 Modeling Framework Configurations


We performed our analysis by creating maximum inundation extent Event Maps produced by three frameworks: the HEC-River Analysis System (HEC-RAS) framework, the AutoRoute framework, and Fathom-US framework. Figure 2 illustrates the inputs for each modeling framework. The proceeding section describes these frameworks in detail. We utilized only observed meteorological and coastal data to ensure that limitations in forecast skill are not present.






**Figure 2: System flow chart for each of the Event Map modelling frameworks that we compare in this study: (a) HEC-RAS modeling**
**framework, (b) AutoRoute modeling framework, and (c) Fathom-US modeling framework.**

In the HEC-RAS framework, the USACE Hydrologic Engineering Center (HEC) – Hydrologic Modeling System (HMS)
version 4.3 model (Hydrologic Modeling System (HEC-HMS): Release Notes, 2018) simulates rainfall-runoff processes
within the watershed. RainVieux radar and gauge derived precipitation data forces the HEC-HMS model (RainVieux, 2022).
The HEC-RAS version 5.0.7 (HEC-RAS River Analysis System: Release Notes, 2019) simulates hydrodynamics conditions




by utilizing one-dimensional (1D) unsteady routing in the main stem of Clear Creek and two-dimensional (2D) diffusive wave routing in the overland and tributaries of Clear Creek. Internal boundary conditions within the HEC-RAS model link HEC-HMS runoff estimates with the HEC-RAS simulation. The HEC-RAS model has a one-way coupling with the coast via downstream boundary conditions along the coast forced with a head value derived from nearby tidal gage readings from NOAA's Tides and Currents dataset (NOAA Tides & Currents, 2021b). LiDAR data obtained from the Texas Natural

Resources Information System (TNRIS) and Houston-Galveston Area Council of Governments (HGAC) provides the HEC-RAS model an approximately 1-meter horizontal resolution terrain (StratMap: Elevation – Lidar, 2021). The Harris County Policy, Criteria, and Procedures Manual (PCPM) provides the 1D roughness coefficients (HFCD, 2018). Land use estimates, derived from Galveston County Appraisal District (GCAD) and Harris County Appraisal District (HCAD) parcel data, estimate roughness coefficients for all 2D areas.

Across the same domain and for the same tropical storm, we develop an AutoRoute estimate of the Event Map that constitutes the AutoRoute framework. We acquire streamflow forcing data from the NWM version 1.2 via Amazon Web Services (NOAA, 2018). The maximum discharge simulated by the NWM then pairs with the associated National Hydrography Dataset Plus (NHDPlus) version 2.0 medium resolution stream reach shapefile (USEPA, 2019a). For topography, we acquire 1/3 arc second (~9 m) horizontal resolution National Elevation Dataset (NED) digital elevation model (DEM) data (Gesch et al., 2002, 2010)

for the study area. The 2016 collection of the National Land Cover Dataset (NLCD, Yang et al., 2018) and literature-derived roughness coefficients as described in Follum et al. (2017, 2020) provide estimates of surface roughness. Because the chosen DEM does not contain bathymetry, we implement the simple bathymetric estimation methodology within AutoRoute (Follum et al., 2020) by using the gage adjusted, Enhanced Runoff Method (EROM) mean annual flows (USEPA, 2020b). The setup of AutoRoute is a representative workflow for implementing a large-scale, steady state hydraulic model for Event Map

development.

    The Fathom-US framework accounts for fluvial, pluvial, and storm surge flooding within one comprehensive framework. Wing et al. (2017; 2019) provide the specifications of the model set up. Observed precipitation data from NOAA's Advanced Hydrologic Prediction service was input into the Fathom-US model to account for pluvial flooding. NWM version 1.2 analysis and assimilation streamflow estimates and USGS National Water Information Service (NWIS) streamflow produce fluvial

flooding. The Fathom-US model simulates interactions between inland and coastal waters by using streamflow data from the combination of the NWM and NWIS and observed water levels from the NOAA Tides and Currents service. The observed NOAA Tides and Currents are input as a downstream boundary condition into the Fathom-US framework at oceanic computation cells, just off-shore from coastal flood defenses (Wing et al., 2019).

### 2.2 Evaluation Methods

We perform two layers of analysis in our assessment to ascertain key differences between each of the three Event Maps. We summarize the analysis of Event Map in the Figure 3 flow chart. The first analysis makes use of U. S. Geological Survey (USGS) high water mark (HWM) data collected following the devastation of Hurricane Harvey (Watson et al., 2018) and


distributed by the USGS Flood Event Viewer (Flood Event Viewer, 2021). The USGS did not produce an estimated inundation map for Clear Creek during Hurricane Harvey, so our comparison focuses on the location and water surface elevation (WSE)

observed at HWMs. We assess locational accuracy for each Event Map by determining the fraction of HWMs that are within the flood extents of the Event Map.

$$Locational\ Accuracy = \ 100 * \frac{N_w}{N},$$  (1)

In Equation 1, $N_w$ is the number of HWMs that are within the flooded extent of each Event Map and N designates the number of HWMs.

Following the methodology outlined by Wing et al. (2021) we assess the estimated WSE from each framework by estimating error and bias.

$$Error = \frac{\sum_1^N |WSE_{mod} - WSE_{obs}|}{N},$$  (2)

$$Bias = \frac{\sum_1^N (WSE_{mod} - WSE_{obs})}{N},$$  (3)

In Equation 2 and Equation 3, WSE$_{mod}$ designates the WSE at the inundated pixel nearest to each HWM location modeled by each Event Map framework, and WSE$_{obs}$ designates the WSE observed at each HWM location.

The second analysis provides a comparison of exposure and consequence estimates from each Event Map. To perform our exposure and consequence analysis, we utilize the Go-consequences model and the National Structural Inventory (NSI) (USACE, 2021a; 2021b; 2021c). The NSI is a point based structural inventory describing structures throughout the United

States. The NSI supports the assessment of consequences of structures resulting from natural and man-made disasters (USACE, 2021c). Go-consequences uses the NSI to compute building damage and population exposure from flooding. Go-consequences uses a water depth estimate at NSI point locations, and uses the default depth-damage functions used within the HEC-Flood Impact Analysis (HEC-FIA) software and assigned by the USACE Economic Guidance Memorandum 04-01 (USACE, 2003). In this instance, our flood damage assessment does not adjust damages to account for brackish water damage (USACE, 2021b).

To visualize the resulting point damage and exposure estimates, we used the point damage locations and their associated dollar damage and building population counts to construct kernel density maps in ArcGIS version 10.8 (Kernel Density, 2022). The kernel density plots can provide a 'hot-spot' analysis to compare to collected Federal Emergency Management Agency (FEMA) flood insurance claim locations (Arctur, 2021). Although total monetary damage to buildings and their contents is difficult to observe following a flood event. Flood insurance claims represent a fraction of the overall damage and may

represent the only spatially explicit observations of monetary flood damage. Shao et al. (2017) summarize that Galveston and Harris County, TX, have rates of flood insurance purchase that fall between 26-50%. Our study domain falls between Galveston and Harris County. Using these flood insurance purchase rates along with the total FEMA flood insurance claims from Harvey constitutes an approximate upper and lower bound of total monetary damage to buildings and contents.



**Figure 3: Flow diagram describing the two pronged evaluation process undertaken to examine the physical differences in each Event Map and the differences in exposure and consequences estimated by each Event Map.**



## 3 Results and Discussion

### 3.1 Simulation Comparison

We first compare the results from the HEC-RAS, AutoRoute, and Fathom-US frameworks to observed HWMs by estimating locational accuracy. HWMs designate locations where floodwater reaches a given location and leaves behind evidence of floodwater presence in the form of mud lines, seed lines, etc. (Koenig et al., 2016). USGS quantifies the uncertainty of the HWM WSE measurements they collect. In our study domain, USGS considers 53% of HWMs in the study area of poor quality, 34% of fair quality, and 13% of good quality. What these qualitative descriptors translate into quantitatively is an average of

± 9 centimeters of uncertainty in the study domain HWM WSEs.

Each models Event Map should contain each HWM. Table 1 is an assessment of locational accuracy for each model under the assumption that the maximum inundation extent should contain the HWM locations. Interestingly, we can see that the Fathom-US model is more accurate at capturing HWM locations within the inundation extent than both the AutoRoute and the HEC-RAS model. This result contradicts our assumption that the HEC-RAS model will be more accurate given the level terrain

resolution and calibration/validation performed upon the model.

**Table 1: Locational accuracy of each modeling framework Event Map based on the number of HWMs within the Event Map flood extent.**

|  | Total Number of High Water Marks | High Water Marks within Event Map | Locational Accuracy |
|---|---|---|---|
| **HEC-RAS** | 56 | 33 | 59% |
| **AutoRoute** | 56 | 13 | 23% |
| **Fathom-US** | 56 | 44 | 88% |






However, expressing model skill in terms of locational accuracy has limited viability, given that the model could inundate the entire study area and achieve 100% accuracy. Indeed, comparing the results from HEC-RAS, AutoRoute, and Fathom-US maximum water surface elevations (WSEs) to observed HWM WSEs reveals a different outcome. Figure 4 illustrates scatterplots comparing each simulation's maximum WSE to HWM WSE observations and Table 2 summarizes the error and bias of each framework. The orange line in all Figure 4 plots is the desired 1:1 relationship between observation and model

results and the hashed line is the line of best fit from a least squares regression analysis. In Figure 4 and Table 2, we see that the HEC-RAS framework produces more precise and accurate WSE estimates than both the AutoRoute and Fathom-US frameworks with points tightly packed along the dashed regression line that align well with a 1:1 line and lower error value. The HEC-RAS frameworks biases toward underestimation with bias of -0.28 MASL. The Fathom-US framework tends to overestimate WSE, with the regression line falling to the right of the 1:1 line and a positive bias of 0.60 MASL. The AutoRoute

framework has a less consistent tendency than the HEC-RAS and Fathom-US frameworks. With only 23% of HWM locations falling with the inundated area, AutoRoute appears to underestimate inundated area. However, the AutoRoute Event Map biases towards significant overestimation with large over predictions illustrated in Figure 4. Injecting each USGS HWM's WSE measurement uncertainty into our error analysis, we find that USGS measurement uncertainty in the HWM WSEs translates into an average of about ±1 cm difference in errors reported in Table 2. Overall, the performance of the HEC-RAS

and Fathom-US frameworks is better that the AutoRoute framework. We certainly expected AutoRoute to underperform in this scenario given the relatively simple numerical scheme and a lack of both pluvial and coastal flooding.


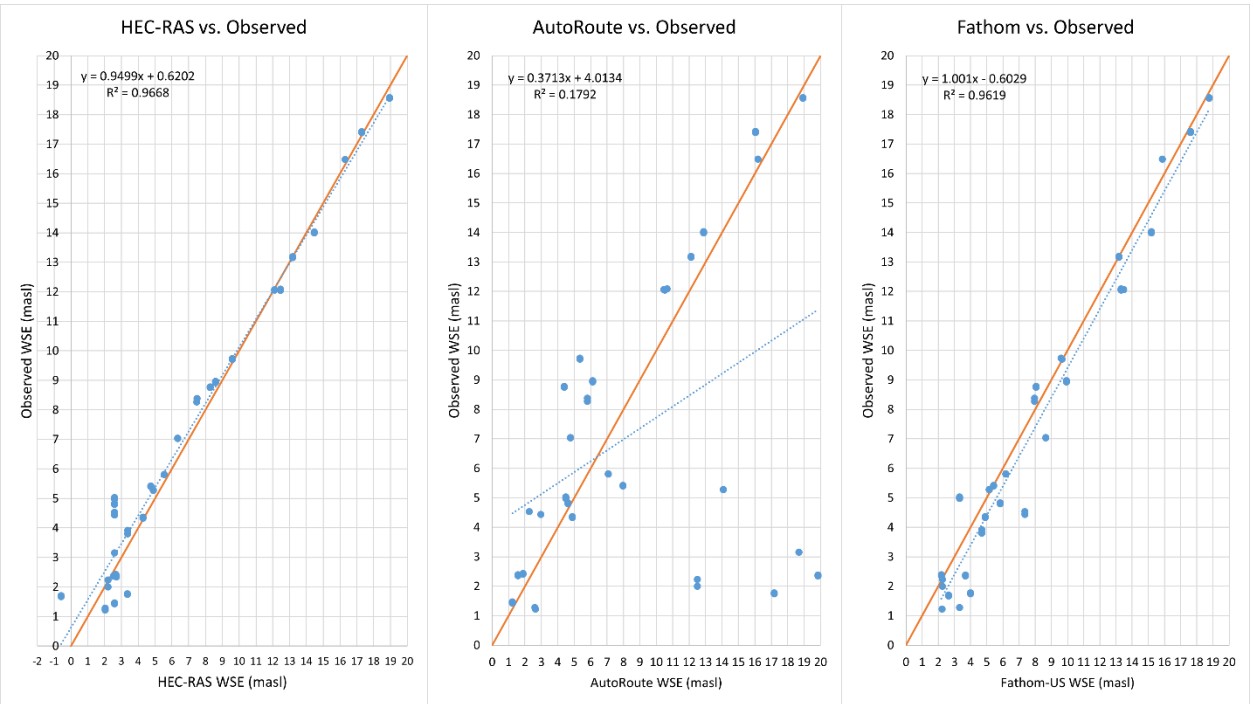

**Figure 4: Scatterplot comparing simulated and observed WSEs for Hurricane Harvey. Here, each blue dot represents an observed HWM location, the orange line represents a 1:1 perfect fit, and the blue dashed line is the line-of-best-fit between the observed and simulated WSE at the HWM locations.**

**Table 2: Error and bias computed for each Event Map framework using observed HWM WSE.**

| Event Map Framework | Error (MASL) | Bias (MASL) |
|---|---|---|
| HEC-RAS | 0.68 | -0.28 |
| AutoRoute | 3.63 | 1.44 |
| Fathom-US | 0.87 | 0.60 |

Figure 5 illustrates the Event Maps created by HEC-RAS, AutoRoute, and Fathom-US along with HWM WSE comparisons with observations for each Event Map. HWMs colored in shades of grey are locations where the Event Map over-predict WSE, those colored red represent under-prediction by each Event Map, and locations colored white are within ± 0.5 MASL for the Event Map. The HEC-RAS and Fathom-US Event Maps are more similar to one another than the AutoRoute Event Map is to either. However, we do see the HEC-RAS framework underestimates WSE in the northeast section of the study area, while the Fathom framework overestimates WSE and estimates greater inundation extents in the northeast than the HEC-RAS framework. The AutoRoute framework underestimates WSE inland and overestimate WSE closer to the coast. Overall, we see that the Event Maps created by each framework are unique to the framework in terms of both model error and overall flood inundation.
**Figure 5: Maps comparing Event Maps from each modeling framework and differences between simulated and observed WSE: (a) HEC-RAS, (b) AutoRoute, and (c) Fathom-US. Each point location is an observed USGS HWM location and the colors represent the magnitude of difference between observed and simulated WSE. The blue region represents the flood inundation depths for each Event Map.**

## 3.2 Causes of Framework Differences and Uncertainty

In general, we see the HEC-RAS, AutoRoute, and Fathom-US frameworks generate different Event Maps and that each is an imperfect representation of reality. The HEC-RAS Event Map appears to underestimate WSE. The Fathom-US Event Map appears to overestimate WSE. The AutoRoute results are a mix of underestimation and overestimation of WSE. As expected, the magnitudes of error is generally higher for the AutoRoute and Fathom-US models than the HEC-RAS results as they are both large scale frameworks. The HEC-RAS and Fathom-US results appear to be a more accurate representation of flooding



that the AutoRoute framework. Here we explore the major drivers of differences and uncertainty amongst the estimated Event Maps.

One of the major differentiations of the AutoRoute framework from the HEC-RAS and Fathom-US frameworks is the missing
coastal component of the Event Map. AutoRoute has proven capable in a variety of inland scenarios (Follum et al., 2017; 2020) and when compared to higher resolution, inland models (Afshari et al., 2018). However, in this instance, it appears that the simplified physics in our AutoRoute simulation do not accommodate the complex physical interactions that occur during this compound coastal flood. In our case study, the AutoRoute Event Map under-predicts WSE but is also prone to large outliers of over-estimation in WSE estimation. As we expected, the HEC-RAS and Fathom-US frameworks outperform the
AutoRoute framework in terms of error.

Unlike AutoRoute, the HEC-RAS and Fathom-US frameworks employ a similar physical fidelity in their respective numerical schemes. The main difference in these simulations is the geographic resolution and vertical accuracy of the DEMs. The HEC-RAS framework uses ~1-meter DEM resolution with an average National Standard for Spatial Data Accuracy (NSSDA) absolute vertical accuracy of about 0.02 meters (AECOM, 2018). The Fathom-US model simulating at around a ~30-meter
DEM resolution has an average NSSDA absolute vertical accuracy of 3.04 meters (Gesch et al., 2014). The greater accuracy and higher resolution of the DEM within the HEC-RAS framework is likely one of the main drivers of the better accuracy and precision within the HEC-RAS Event Map as compared to the Fathom-US Event Map. The Fathom-US DEM has a lower vertical accuracy, which is likely one of the major drives in the difference between the Fathom-US and HEC-RAS Event Maps. The horizontal resolution of the DEM will also play a role in the accuracy of Event Maps, particularly when the domain is an
urban catchment, unless small-scale influences on the hydraulic conditions are present in the DEM (Wing et al., 2017; 2019; Domeneghetti et al., 2021). We consider our study area a densely urbanized watershed (Bass and Bedient, 2018).

One of the more apparent differences between the HEC-RAS and Fathom-US frameworks is the omission of HEC-HMS internal boundary conditions in the northeast corner of the watershed. This region is Armand Bayou. The HEC-HMS runoff for Armand Bayou enters the HEC-RAS model near the pour point of Armand Bayou and not in a distributed manner
throughout the Armand Bayou watershed. The Armand Bayou watershed is under analysis in a separate study and the Clear Creek HEC-RAS model only considers the total runoff coming into the Clear Creek domain from Armand Bayou. When developing an Event Map with different frameworks, the user must understand the assumptions made by the modeler. In this instance, the application of runoff from Armand Bayou enters the HEC-RAS framework. However, because the runoff is not applied in a distributed manner throughout the watershed, an under representation of modeled inundation occurs upstream of
Armand Bayou's pour point, effectively removing pluvial and fluvial flooding from the region.

The Fathom-US framework is the only framework we consider that explicitly accounts for pluvial flooding from precipitation. The Fathom-US framework does this by performing a rain-on-grid simulation that relates local soils and land use to infiltration capacity and drainage design standards (Sampson et al., 2013; Wing et al., 2019). The AutoRoute framework is exclusively riverine, simulating inundation by converting the maximum streamflow from the NWM into a flood inundation extent. The
HEC-RAS framework inserts runoff from the HEC-HMS simulation into the HEC-RAS modeling domain via internal


boundary conditions that represent runoff into Clear Creek's tributaries. The disparities in how each framework does or does not account for pluvial flooding are likely to be one of the drivers of the greater inundation extents present in the Fathom-US Event Map. The lack of pluvial flooding within an Event Map is an inherent limitation in the current AutoRoute framework. However, in the HEC-RAS framework, the limitation is not inherent within the framework but is a decision made by the modeler. HEC-RAS Version 5.0.7 can perform rain-on-grid simulations. However, HEC-RAS version 5.0.7 lacks the ability to apply distributed rainfall across the model domain, instead applying a uniform value across the entire simulation domain. Further, HEC-RAS version 5.0.7 is unable to account for infiltration, converting all rainfall into subsequent runoff (HEC-RAS River Analysis System: Release Notes, 2019). Thus, the pluvial component of our Event Map is both a limitation in some frameworks and a modeler's decision in other cases. The explicit exclusion of pluvial flooding appears to have some influence in our Event Map results.

### 3.3 Implications of Model Differences

As expected, each of the three modeling frameworks we consider estimate different Event Maps. The differences in Event Maps translate into different estimates of consequences and exposure. Table 3 summarizes the consequence and exposure differences. We see higher WSE, full inclusion of Armand Bayou in the northeast section of the study domain, and explicit inclusion of pluvial flooding in the Fathom-US model translates into larger consequence and exposure estimates. The Fathom-US Event Map estimates that floodwater from Harvey inundated approximately 39% of all buildings in the study domain while HEC-RAS and AutoRoute Event Maps estimate 10% and 3% of all buildings in the study domain were inundated with flood waters, respectively. Interestingly, there is not a general trend of increasing estimates of exposure that lead to increases in our estimates of dollar damage. The HEC-RAS Event Map predicts flood exposure to 13,002 more structures than the AutoRoute Event Map. However, the AutoRoute Event Map estimates $0.2 billion more in total monetary damage than the HEC-RAS Event Map. Exploring this result, we find that at buildings structures where both the HEC-RAS and AutoRoute Event Maps predict damage, go-consequences predicted approximately $0.3 billion more in damages for AutoRoute, damages where only HEC-RAS and not AutoRoute predicted damage is approximately $0.7 billion, and damages where only AutoRoute and not HEC-RAS predicted damages is about $0.5 billion. AutoRoute estimates $0.3 billion more at the same locations inundated by HEC-RAS, indicating that AutoRoute depth estimates are higher at these locations and estimate more damage. At locations where either AutoRoute and not HEC-RAS or HEC-RAS and not AutoRoute predict flooding, average depths are 3.8 m and 1.1 m, respectively. Thus, the AutoRoute framework appears to estimate higher total damage values while exposing fewer buildings than those produced by the HEC-RAS framework because of a bias towards greater water depth estimates at those locations. These results indicate that the differences in each Event Map produce different estimates of both exposure and consequences.





**Table 3: Consequence and exposure estimates for Clear Creek during Hurricane Harvey estimated using each Event Map and the**
320  **go-consequences software.**

|  | Estimated Number of Structures Impacted | Estimated Total Depreciated Damage (Structure and Content Values 2018 Dollars) | Total Exposed Population at Night (under age 65) | Total Exposed Population at Night (over age 65) | Total Exposed Population During Daytime (under age 65) | Total Exposed Population During Daytime (over age 65) |
|---|---|---|---|---|---|---|
| **HEC-RAS** | 19,281 | $0.7 Billion | 50,228 | 6,000 | 57,960 | 5,585 |
| **AutoRoute** | 6,279 | $0.9 Billion | 14,948 | 1,884 | 9,593 | 1,659 |
| **Fathom-US** | 72,601 | $3.3 Billion | 193,761 | 22,513 | 147,605 | 20,051 |

We use the locations of the buildings impacted, the damage to those building, and the number of people within those buildings from each Event Map go-consequences analysis to construct a kernel density map (Figure 6) where we see a spatial pattern that matches the tabular values in Table 3.  The HEC-RAS framework estimates that the highest density of impact will be in

325  the western and southern portions of the study domain. As stated before, the HEC-RAS framework omits distributed internal boundary conditions in Armand Bayou watershed in the northeast portion of the study area, due to the modeling assumptions. The AutoRoute framework estimates that the highest density of impacts will occur in pockets throughout the study domain. The Fathom-US framework mimics the spatial pattern of the HEC-RAS framework but broadens estimates of impact throughout the entire study domain and in particular in the northeast section that the HEC-RAS framework omits. Overall, the

330  densities portrayed in Figure 6 match well will the magnitudes of consequences and exposure portrayed in Table 3. Thus, exposure estimates produced by each Event Map differ both in their magnitude and in spatial pattern.





**Figure 6: Kernel density maps of buildings, damages, and people per square kilometer impacted by each Event Map: (a) HEC-RAS (b) AutoRoute (c) Fathom-US. The kernel density maps derive from go-consequences point output for each Event Map modeling framework.**





The Event Maps produced by each framework are different in terms of their spatial and physical composition and each estimates different consequences and exposures to the floodwaters. We may assume that the Event Map produced by HEC-RAS is the most accurate given the better fit between observed and simulated WSE (Figure 4 and Table 2). However, the HEC-RAS framework is not without error, has a lower locational accuracy than the Fathom-US framework (Table 1), and does not intend to represent flood inundation in the northeast section of the study region (Armand Bayou). Furthermore, as we compare FEMA flood insurance claim locations from Hurricane Harvey (Arctur, 2021) to each Event Map, we find evidence that the HEC-RAS framework Event Map is indeed excluding flooding in the northeast portion of the study area. Figure 7 compares the location of FEMA insurance claims for structures in the AOI and the estimate of buildings per square kilometer from Figure 6. Figure 7 illustrates that both the HEC-RAS and Fathom-US frameworks do well in identifying hotspots of buildings impacted by flooding in the western and southern portions of the study area. However, the HEC-RAS framework does exclude impacted areas in the northeast portion of the study area, while the Fathom-US framework correctly identifies those locations. The AutoRoute Event Map does not appear to perform well at identifying the spatial pattern of impacted buildings.


**Figure 7: Comparison of the location of FEMA flood insurance claims and kernel density maps developed using go-consequences results for each Event Map: (a) HEC-RAS (b) AutoRoute (c) Fathom-US.**

When we calculate the proportion of FEMA insurance claims falling within each Event Map's flood inundation extent (Table 4), we see that none of the frameworks captures all FEMA claims and the pattern echoes the quantitative pattern of HWM data in Table 1. However, if we sum all FEMA claims that fall within at least one of the Event Maps we capture a slightly greater portion of FEMA claims. Thus, our case study may further illustrate the importance of a multi-model, probabilistic approach to Event Map creation that can convey uncertainty in the chosen Event Map framework (e.g., HEC-RAS, AutoRoute, Fathom-

US). Deterministic Event Maps, considering only one source of inundation mapping, do not convey to the decision maker the cascading uncertainty in the methods, tools, and data that create the Event Map (Merwade et al., 2008). NOAA's National Hurricane Center (NHC) has worked extensively with stakeholders to develop products that assist with decision-making.





Stakeholder groups have tended to favor probabilistic maps for the maps ease of understanding and usefulness (NOAA, 2013).

Thus, an Event Map produced for a given compound coastal flood should likely follow a similar convention.


**Table 4: Proportion of FEMA insurance claims within the Event Map for a combination of all three Event Map modeling frameworks.**

| Model | Proportion of FEMA flood claims within Event Map Flooded Area |
|---|---|
| **HEC-RAS** | 56% |
| **AutoRoute** | 6% |
| **Fathom-US** | 79% |
| **All Frameworks Combined** | 86% |

In 2018 dollars, residents of the study area made roughly $898 million in total FEMA flood insurance claims in the wake of

Hurricane Harvey. Thus, using the assumption that between 26-50% of residents in our study domain possess flood insurance, and dividing the total flood insurance claims by these proportions, we estimate between $1.2-3.6 billion of total flood damage to structures and contents resulted because of Harvey in our study domain. These values bounds roughly correspond with the $0.7-3.3 billion estimated by each Event Map. Thus, we surmise that our total damage estimates we produce roughly align with what occurred in reality during Hurricane Harvey.

**3.3 How to Improve Event Map Creation Techniques**

Efforts are ongoing to coordinate Event Map creation at the federal level. The three frameworks discussed in our study are not the only techniques available to create Event Maps during flood events. As previously mentioned, the NWC produces HAND-derived Event Maps using the NWM (Viterbo et al., 2020). The U. S. Department of Homeland Security (DHS) contracts with the Pacific Northwest National Lab (PNNL) to construct Event Maps with the Rapid Infrastructure Flood Tool (RIFT) model

(Judi et al., 2010; PNNL flood modeling helps DHS during busy hurricane season, 2017; Li et al., 2019). NOAA's NWS and the USGS host multiple flood mapping libraries (Inundation Mapping Locations, 2022; Flood Event Viewer, 2021). There are likely other entities capable of producing Event Maps throughout the country. Our case study highlights how the three Event Map frameworks we consider are different, imperfect, and can lead to different estimates of flood exposure and consequences. Thus, there is a need to reconcile and adjudicate multiple Event Maps to ensure consistency in decision-making efforts during

flood events. In response to this need, the IWRSS consortium has set about operational plans for coordinating Event Map production through the integrated Flood Inundation Mapping (iFIM) effort (Gutenson, 2020). The iFIM group confers before, during, and after major flood events in order to promote awareness of the various Event Map creation efforts. The iFIM effort is in its infancy, gathering together to understand the where and when of Event Map production. However, this is a necessary first step in building cohesion in developing appropriate Event Maps. In our current context, the iFIM group would have been





aware that the HEC-RAS framework should not be representative of the northeast section of the study domain and that the AutoRoute framework generally performs poorly in low gradient coastal watersheds. This adjudication process would have likely led to the iFIM group promoting the Fathom-US framework for use in the northeast section of the study region and the HEC-RAS framework in the rest of the study area as the most appropriate Event Map.

To empower the iFIM group, additional steps to enable interoperability and sharing of maps across multiple levels and divisions
of government will also be necessary. From a practical perspective, this means developing data services to share amongst the different agencies. NOAA's NWS, USACE, and USGS all provide access to Event Maps through geographic information systems (GIS) services. The next step will be the engagement of other Federal entities and those that fall outside of the Federal agencies. Simply exposing these Event Maps as GIS services and allowing the iFIM group to import them within a common operating picture will empower the Event Map adjudication and promotion process.

The iFIM intends to promote the most appropriate Event Map for a given flood event and location. However, as we have seen with this case study of Clear Creek during Hurricane Harvey, a deterministic Event Map can be problematic for compound coastal flooding given that all chosen modeling frameworks produce an imperfect assessment of reality. As Table 4 displays, our combination of all three Event Maps encompasses a greater proportion of FEMA flood claims than one location alone. Thus, we have some initial evidence to suggest that the delivery of a multi-model Event Map should be the preferred
methodology to Event Map delivery.

However, a chosen Event Map framework highlights only one aspect of the uncertainty within Event Map creation. This assessment has not considered the uncertainty associated with the use of numerical weather prediction (NWP) models. Even with gains in NWP forecast skill, the use of ensemble prediction remains key to understanding the uncertainty when predicting chaotic weather systems. Ensemble prediction entails the perturbation of initial conditions and model numerical schemes to
create a range of possible meteorological conditions (Palmer, 2017). Thus, the delivery of an ensemble, multi-model probabilistic Event Map should be the preferred methodology to deliver an Event Map in order to convey uncertainty to decision makers. Figure 8 represents a hypothetical inland centric, forecast system where n number of NWP hydrometeorlogical ensemble forecasts force a tide and storm surge model and the hydrometeorlogical and tides and surge ensembles force each Event Map framework. The result of such a system would be a multi-model ensemble based probabilistic
Event Map, similar to that proposed by Zarzar et al. (2018). In general, expansion of the full expression of knowledge uncertainties, extending beyond model selection and NWP forcing into areas such as coefficient determination for hydraulic structures, will generally benefit the portrayal of flood risk in Event Maps.

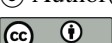


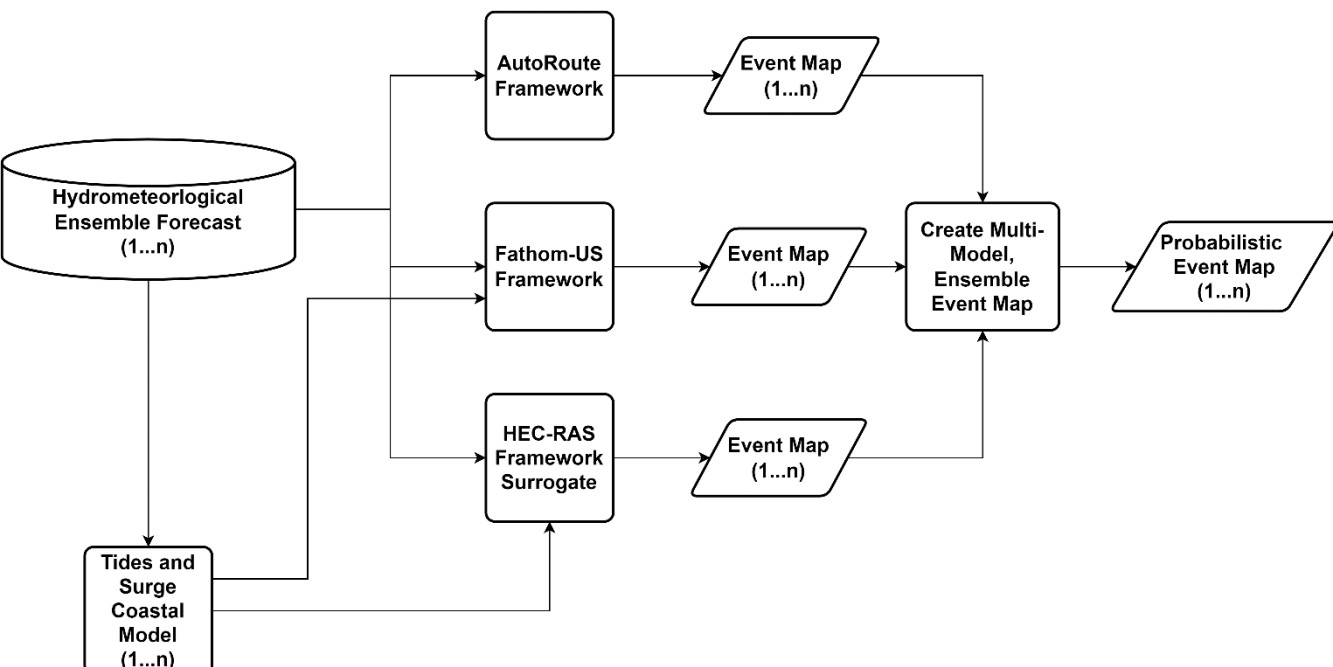

**Figure 8: Hypothetical multi-model, hydrometeorological forecast with the AutoRoute and Fathom-US frameworks supplemented with a surrogate model of the HEC-RAS framework to generate a probabilistic Event Map.**

Two of the large-scale frameworks (Fathom-US and AutoRoute) we employ here have the potential to generate timely probabilistic Event Maps using hydrometeorological ensemble forecasts (Wing et al., 2019). However, to take advantage of hydrometeorological ensemble forecasts within local-scale frameworks, such as our HEC-RAS example, we must effectively reduce model runtime. The setup and runtime of these local frameworks may affect the timeliness of Event Map creation, which is crucial during emergency operations (Follum et al., 2017; Longenecker et al., 2020; Gutenson et al., 2021). However, local scale frameworks offer high fidelity, high-resolution products that can improve a probabilistic Event Map. One means to reduce model runtime for local scale models is to develop and train surrogate models that can dramatically reduce the computational runtime of high fidelity, high-resolution modeling while delivering similar results (Zahura et al., 2020; Contreras et al., 2020; Kyprioti et al., 2021). In fact, Bass and Bedient (2018) have already developed such a surrogate modeling approach to create an Event Map within our study area that loosely couples inland and coastal models, forcing both with a full range of potential tropical cyclone characteristics. However, accurately training surrogate models for compound hazards is not trivial, given the need to expose the surrogate technique to numerous pre-existing simulations that account for the multitude of physical interactions, initial conditions, etc. that expand beyond tropical cyclone forcing.

Inevitably, improvements in numerical schemes and input data should provide improvements in Event Map creation. In their review of the literature, Santiago-Collazo et al. (2019) determine that 96% of the literature they analyse presents compound coastal flood inundation modeling strategies employ one way coupling. By one-way coupling, we mean were outputs from one model (e.g., inland) are fed into another model (e.g., coastal) by way of internal or external boundary conditions and no





feedback occurs between the coupled models. The HEC-RAS and Fathom-US frameworks discussed here are examples of
one-way coupling strategies as the models insert coastal surge into both frameworks via downstream head boundary conditions.
Santiago-Collazo et al. (2019) advocate for the use of more robust coupling strategies to account for the complex interaction
between inland runoff and storm surge; such as loosely-coupled, tightly-coupled, or fully-coupled modeling strategies. Further,
Event Map improvement will undoubtedly occur as improvements to the widespread availability of critical input datasets
occur. For instance, the USGS collection of improved DEM data is steadily decreasing vertical and relative DEM errors (Gesch
et al., 2014).

## 4 Conclusions

In this manuscript, we compare three different Event Map creation frameworks for a small coastal watershed, Clear Creek,
near Houston, Texas during Hurricane Harvey. These frameworks are the HEC-RAS framework, the AutoRoute framework,
and the Fathom-US framework.

We estimate the maximum flood inundation raster from each framework, considering this our Event Map (IWRSS, 2013). We
then compare each framework's Event Map to USGS HWMs in two ways. First, we assess whether the Event Map contains
each HWM within the estimated flood extent. Second, we compare observed WSE from the USGS HWM to estimated WSE
in the Event Map. Our analysis indicates that Event Map accuracy can vary based upon either of these assessments. The
Fathom-US framework contains the most HWMs but also tends to overestimate WSE. The HEC-RAS framework contains
less HWMs but also tends to have relatively more accurate WSE. The AutoRoute framework is the least accurate of the three,
appears to underestimate flood extent, and highlights how simplified flood inundation mapping methods are not ideal for
representing compound coastal flooding. Our analysis illustrates that no one Event Map is infallible and is subject to the
uncertainties present in the model's numerical scheme, the model inputs (e.g., terrain), and the model's configuration.

We the estimate the exposure and consequences of each Event Map using the NSI and go-consequences. We find quantitative
and spatial differences in the exposure and consequences produced by each Event Map. The differences we find between each
Event Map further illustrate why a singular, deterministic Event Map is not preferable. We compare our exposure and
consequence estimates to the locations of FEMA flood claims and use FEMA damage claims totals to estimate a total damage.
Visually (Figure 7) and numerically, the comparison of simulated exposure and consequence estimates compare favorably to
our approximate observations. The results lend credence to our ability to utilize accurate Event Maps, the NSI, and go-
consequences and produce a relatively accurate exposure and consequence assessment for a flood event. Thus, the combination
may be a useful tool set for evaluating the impacts of floods before, during, and after they happen.

Our study highlights the need to rectify and adjudicate the various Event Maps created during flood events. In response to this
need, IWRSS formed the iFIM to perform interagency comparison and consolidation of Event Maps. GIS web services
empower the iFIM and adding additional Event Maps to the iFIM common operating picture will improve the Event Map
selection and discovery process.


Large-scale Event Map creation techniques, such as AutoRoute and Fathom-US may be capable of operating in real-time during flood events. To develop Event Maps properly for compound floods and beyond, future research should focus on means to reduce runtime in local-scale models that offer high-fidelity numerical schemes and high-resolution input data. Surrogate modeling may offer such an approach but the difficulties in training a multivariate surrogate model are not trivial. Decreased

runtimes may offer the ability to instantiate multiple model simulations while not compromising model fidelity. This would make possible probabilistic Event Maps for compound coastal floods that capitalize on the fidelity and resolution of local-scale models.

### Author contribution

JLG and AAT conceptualized the study. JLG designed the study and conducted the experiments. JLG drafted the manuscript.

MSI and OEJW assisted with data collection and preparation. WPL and COH assisted with model set up and implementation. MSI, OEJW, MDW, and TCM provided comments and feedback on the manuscript draft.

### Data availability

The presented HWM data are accessible on the USGS Flood Event Viewer (Flood Event Viewer, 2021).

### Competing interests

The authors declare that they have no conflict of interest.

### Acknowledgements

This project was funded by U.S. Army Engineer Research and Development Center (ERDC) Collaborative Research between ERDC's Coastal & Hydraulics Laboratory and NOAA's National Water Center and U.S. Army Corps of Engineers, National Regional Sediment Management Program. The authors would like to thank Russell Blessing at the Texas A&M – Galveston

Institute for a Disaster Resilient Texas (IDRT) for providing total estimates of flood insurance claims for use in our study. We appreciate all input that our reviewers will provide.

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
