# Peer review of "Comparison of Estimated Flood Exposure and Consequences Generated by Different Event-Based Inland Flood Inundation Maps"

_Natural Hazards and Earth System Sciences, 2022_

## Referee Comment (RC2)

**Summary**

In this study, the authors compare three modeling frameworks for mapping inundation extent and flood depth in Clear Creek watershed, a tributary of Galveston Bay in Texas, U.S. They evaluate the performance of AutoRoute, HEC-RAS, and Fathom-US frameworks against in-situ USGS high-water marks of Hurricane Harvey, a well-studied compound flood event in 2017. Also, the authors estimate flood exposure, consequences, and damages to buildings using available data from FEMA. It is shown that both HEC-RAS and Fathom-US outperform AutoRoute due to inherent limitations of the latter framework to simulate flooding in low-lying areas. Although Fathom-US and HEC-RAS achieve high location accuracies and low error and bias, they present some discrepancies regarding the evaluation metrics. The authors suggest an ensemble, multi-model probabilistic methodology to leverage these frameworks and provide more accurate flood maps as discussed in similar studies.

**Major comments**

This study presents an inter-model comparison with a practical application in terms of flood exposure and damage assessments, but it does not provide essential information for doing so. In contrast to Fathom-US (Wing et al., 2017, 2019), there is no evidence of model calibration and validation of both AutoRoute and HEC-RAS models for the study area. If the goal is to evaluate model's performance, then input data, forcing, mesh extent, and grid resolution should be identical among the frameworks. This compromises not only the validity of the results, but the analyses presented throughout the manuscript. I suggest the authors to consult or follow other studies that provide guidelines for model comparison (Shustikova et al., 2019; Muñoz et al., 2021; Afshari et al., 2018; Liu et al., 2018).

The authors investigate the performance of the frameworks knowing beforehand that AutoRoute is not suitable in coastal areas (Line 63 in the Introduction). This rises concern about the usefulness of a low-skill model in this study. If the authors want to consider *steady-state* models like AutoRoute (or HAND) in the model comparison, I suggest to follow the approach of Jafarzadegan et al., (2022) to enhance model simulations via hydrogeomorphic classifiers.

**Minor comments**

L16: 'Event maps' is too generic for referring to flood inundation maps. 'Event maps' are also used to describe the modeling framework making the manuscript difficult to follow in some sections.

L20: Are you talking about modeling frameworks or flood inundation maps? How can event maps be physically different?

L26: Do you mean flood emergency response instead of flood fights?

L28: We find that the **modeling frameworks** are much different physically…

L43: Typo. Event Maps help emergency managers…

L63: HAND can be adapted to simulate coastal flooding in low-lying areas. See Jafarzadegan et al., (2022).

L88: I can anticipate that you will find substantial differences based on the DEM resolution and forcing data you have chosen for each framework.

L98: Details of the hydrologic and hydraulic modeling are missing. For example, what is the grid size for the 2D component?

L120: Diffusive wave is a simplified version of the shallow water equations. Given the nonlinearities and complexities arising in compound coastal flooding, the complete set of equations (SWL) available in HEC-RAS should be used. This might lead to a better accuracy of the HEC-RAS in terms of inundation extent and flood depth.

L123: What are those mysterious downstream boundary conditions? Figure 1 should include the location of those boundaries for the three modeling frameworks.

L129: Are roughness values calibrated afterwards? These initial 1D and 2D roughness values are event-specific and have to be tuned for future flood events.

L134: Would it not be better to consider the 1-m DEM and so avoid inaccuracies due to DEM resolution? Previously, you suggest using observed meteorological data to avoid limitations in forecast skill...

L150: Evaluation of simulated time series is very informative but missing in this study (e.g., timing and magnitude of peak water level). I strongly suggest assessing model's performance based on time series of available USGS (#08077637) and NOAA stations.

L183: What are the upper and lower bounds?

L185. Typo in the diagram. "Create Kernel density maps".

L197: Diffusion wave does not solve the full mass balance and momentum equations and therefore might have influenced flood inundation extent and depth. In addition, the 1D portion of the model cannot provide 2D flood maps and consequently miss nearby high water marks.

L200. I cannot find the calibration and validation process in this manuscript. The same holds for AutoRoute model.

L218: USGS high-water marks are referenced with respect to NAVD88. There may be uncertainties added in the NAVD88 to MSL conversion process. How did the authors conduct the datum conversion? What is the vertical datum of the DEMs?

L226: … and the steady state assumption. Also, AutoRoute is only forced by streamflow ignoring the contribution of coastal water level (e.g., storm-tide) to compound flooding (Figure 2).

L228: Figure 4. Text size is too small.

L412: I agree that ensemble modeling is the way to go for better compound flood assessments. Nevertheless, I consider Figure 8 unnecessary in this study as you are not actually following this

approach for simulating compound flooding due to Hurricane Harvey. A descriptive text is enough for future work in this regard.

**References**

Afshari, S., Tavakoly, A. A., Rajib, M. A., Zheng, X., Follum, M. L., Omranian, E., and Fekete, B. M.: Comparison of new generation low-complexity flood inundation mapping tools with a hydrodynamic model, Journal of Hydrology, 556, 539–556, https://doi.org/10.1016/j.jhydrol.2017.11.036, 2018.

Jafarzadegan, K., Muñoz, D. F., Moftakhari, H., Gutenson, J. L., Savant, G., and Moradkhani, H.: Real-time coastal flood hazard assessment using DEM-based hydrogeomorphic classifiers, 22, 1419–1435, https://doi.org/10.5194/nhess-22-1419-2022, 2022.

Liu, Z., Merwade, V., and Jafarzadegan, K.: Investigating the role of model structure and surface roughness in generating flood inundation extents using one- and two-dimensional hydraulic models, 0, e12347, https://doi.org/10.1111/jfr3.12347, 2018.

Muñoz, D. F., Yin, D., Bakhtyar, R., Moftakhari, H., Xue, Z., Mandli, K., and Ferreira, C.: Inter-Model Comparison of Delft3D-FM and 2D HEC-RAS for Total Water Level Prediction in Coastal to Inland Transition Zones, n/a, https://doi.org/10.1111/1752-1688.12952, 2021.

Shustikova, I., Domeneghetti, A., Neal, J. C., Bates, P., and Castellarin, A.: Comparing 2D capabilities of HEC-RAS and LISFLOOD-FP on complex topography, 64, 1769–1782, https://doi.org/10.1080/02626667.2019.1671982, 2019.

Wing, O. E. J., Bates, P. D., Sampson, C. C., Smith, A. M., Johnson, K. A., and Erickson, T. A.: Validation of a 30 m resolution flood hazard model of the conterminous United States, 53, 7968–7986, https://doi.org/10.1002/2017WR020917, 2017.

Wing, O. E. J., Sampson, C. C., Bates, P. D., Quinn, N., Smith, A. M., and Neal, J. C.: A flood inundation forecast of Hurricane Harvey using a continental-scale 2D hydrodynamic model, Journal of Hydrology X, 4, 100039, https://doi.org/10.1016/j.hydroa.2019.100039, 2019.

---

## Author Response (AR1)

Hi Kai,

Thank you for taking the time to review and find reviewers for our manuscript! We are grateful for your consideration of our manuscript for NHESS. The reviewer's comments and suggestions have certainly improved the manuscript.

The focus of our revision has been on ensuring that we effectively communicate the objectives of our study to the reader. To improve the communication of our study's objectives, we not only revise the text but also a revise the title. The focus of the manuscript should not be on comparing flood inundation mapping (FIM) frameworks and attempting to grade them. Rather, we want to explore if spatial differences in the FIMs created different exposure and consequences estimates. This our attempt at determining what are the downstream ramifications of differences in an event-based FIM. We think that our study offers proof that a centralized means of vetting and adjudicating FIM during flood events should occur and that a single FIM should not be the sole means of communicating flood extents. We think that the revised manuscript better reflects the studies objectives.

Below, please find our point-by-point response to the reviewer's comments and suggestions.

If you require any additional information, please do not hesitate to let us know.

Thanks,

Joseph Gutenson

**RC1**

The authors have produced good work in comparing inundation models and some discrepancies between them in representing flood inundation on maps for compound coastal flooding. Some revision should occur before publication to improve the manuscript's clarity and organization. Generally, I agree that Event Maps should be vetted, particularly in cases of compound flooding or the broader case of many inundation mapmakers producing flood estimates with differing data sources for a flood event; however, the three models appearing in this paper have very different purposes that users of the inundation estimates would likely know about—so the paper is more a demonstration that the models produce different flood footprints. Why not use three coastal flood models and include Adcirc or RIFT, among other models designed for the hydrodynamics of compound coastal flooding? If these models are not available for use in this study, please state so. The point is well taken that any one map could be wrong because it omits variables or flood sources, but the authors do not show how the vetting framework proposed would apply to official products, like, for example, the NHC hurricane storm surge inundation graphic. In forecasting, NHC uses numerous models and real-time parameterization decisions in the production of the hurricane storm surge inundation graphic, so it seems there is something to highlight or learn from NHC's model review and communication processes. I've provided some more detailed comments below and would be happy to assist the authors for further review to get this paper ready for publication. Again, it's good work, but some clarifications are needed.

==Response: We appreciate the reviewer's time in performing this thorough review! We think that our clarification and refinement of the manuscript's scope should alleviate your concerns.==

==We certainly agree that the National Hurricane Center (NHC) flood inundation products, along with others such as the Coastal Emergency Risks Assessment (CERA) (https://cera.coastalrisk.live/) should be included in the vetting and adjudication process for the most appropriate flood inundation maps for a given time and location on the coast. However, we think that their omission==

**in this manuscript does not detract from the focus of the manuscript. The focus of the manuscript is to evaluate a sample of flood inundation maps that could be used during a flood event, evaluate if each flood inundation map has a different spatial composition, determine if each spatial composition is imperfect, and determine if the differences and imperfections lead to different estimates of exposure and consequences (i.e., in other others are their differences impactful). The three flood inundation mapping frameworks we select could indeed be deployed as flood inundation guidance during a compound flood event. If the user is a subject matter expert, they will likely be able to discern key differences and employ the most correct map. However, if the user is an emergency manager or the public at large, knowing which map to use can be difficult, making the vetting process critical given that this manuscript provides evidence that each flood map can lead to different conclusions on the spatial pattern and severity of a floods impacts. It's likely an NHC, CERA, RIFT or other flood inundation product would be different than the flood inundation maps we sample but these differences are not necessary to prove the point of the manuscript and all will still have imperfections. We have ensured that the revised manuscript and title of the manuscript better defines the scope of this research and addresses the central tenets of this manuscript appropriately. We have also evaluated the NHC's process and incorporated those into the Section 3.3 discussion.**

Line 45: assumes that sources are authoritative and/or unknown, whereas event maps are typically sourced according to authority as in lines 39-43

**Response: Good point, we now offer clarification here that event maps are authoritative but that non-authoritative maps can and do occur during flood events.**

Lines 71-76 present a conclusion/appears out of order

**Response: We added a concluding statement to the end of this paragraph for clarification.**

Line 88: passive tense

**Response: We made the necessary change to an active tense.**

Figure 2: graphic introduces use of different DEMs and resolutions, introduces hindcast, introduces multiple unexplained acronyms and data sources (are these public?), national water model uses a land surface (NOAA) whereas auto route uses DEM—are these post processes? Do these inconsistencies in land models relate to the differences in inundation maps and/or accuracies?

**Response: The acronyms and data sources in Figure 2 should be introduced in the proceeding text (lines 115-149). We use the acronyms in Figure 2 to simplify the graphic in Figure 2. All data are public except the RainVieux precipitation product used in the HEC-RAS modeling framework. We ensured that we define all acronyms in the proceeding text.**

**We discuss how some of these differences lead to different flood inundation maps (see Section 3.2).**

Line 126: inconsistent units (meter vs arc second in figure 2)

**Response: We will correct this in Figure 2 and in the text of the manuscript.**

Lines 120-149: streamflow data appears to be a consistent variable across the frameworks whereas elevation, roughness coefficients, and bathymetry appear differently—the fathom framework appears well cited/situated in literature but auto route and HEC RAS approaches are less clear

**Response: The HEC-RAS approach is a typical methodology used by U.S. Army Corps of Engineers (USACE) Districts to develop flood inundation mapping frameworks for design purposes. The AutoRoute approach is available in the literature (Follum et al., 2017; Follum et al., 2020). We have clarified these aspects in the text.**

Lines 150-160: comparison with HWM, here and more broadly, could be problematic given different elevation sources and other spatial/vertical corrections

**Response: Any form of evaluation of a flood map will have problematic components because of limited observations and the need to approximate the observed data. For example, the basis of comparing simulated and observed flood inundation extents will use an approximate observed flood inundation extent that uses an elevation source and HWM data or from remotely sensed observations, leading to an approximate flood extent observation. Though the reviewer is correct in this assertion, there will be no perfect comparison because of gaps in the observation data.**

Lines 169-172: Recommend clarifying that the NSI includes population estimates from Census/ACS—it was not clear that population is included in NSI

**Response: We have included this detail in the manuscript.**

Line 178-180: awkward phrasing/qualifier to damage estimates; insurance uptake is a separate but interesting issue—does lower uptake relate to poorer damage estimation? Would you have a better estimate of flood damage bounding if uptake rates increased to 100%?

**Response: Direct comparison between NSI/go-consequence estimates and observations is problematic for a number of reasons. First, personally identifiable information (PII) limitations negate FEMA from sharing disaggregated flood insurance claims with the authors. Second, flood insurance uptake is approximately 25-100% within our study area, varying significantly by county (Shao et al., 2017) and thus, flood insurance claims are likely unrepresentative of total flood damage from Hurricane Harvey. However, even with 100% insurance uptake, matching point observations of flood damage reported in flood insurance claims with point NSI/go-consequence point estimates of flood damage is still problematic because the NSI does not necessarily have attributes, such as structure value, that match the building's insurance policy coverage. Thus for flood insurance claims, the coverage is truncated on the lower end by deductibles (thus losses are not recorded because no claim is made) and on the upper end by policy caps (thus losses in excess of the policy may be truncated to the payout rather than the actual loss). We have incorporated this discussion to the manuscript.**

Figure 3: kernel typo; why is depreciation mentioned (seems more a benefit-cost assessment than a damage assessment)? What is the resolution of the kernel density analysis? Does that represent up/downscaling?

**Response: We have removed the reference to depreciation because we have removed the total damage comparison from the manuscript.**

**The kernel density analysis uses a 1 km search radius and outputs as a 1 km resolution raster. The kernel density analysis provides a visual means to compare our exposure and consequence estimates to FEMA insurance claims, as a direct 1:1 comparison is not possible. We have included these details in the revised manuscript.**

Lines 190-200: are there HWM from Harris County Flood Control District that might supplement the analysis? I understand HCFCD to have collected this data, though it may not be publicly available. The qualitative descriptors for USGS HWM typically refer to whether the HWM itself is recognizable and is not necessarily a description of elevation differences or potential height uncertainty. Further, sources of flooding leaving the HWM is typically noted in USGS data, so it would be appropriate to describe flood sources (are all the HWM coastal, riverine, or compound? Any ponding or other disconnected flooding? Similar comments for Section 3.2).

Response: The inclusion of additional HWM data will not offer additional data to enhance the conclusions of this manuscript given that the primary focus of the study was not to analyze the performance of each flood inundation mapping framework. The focus of analyzing HWM data in this manuscript was to describe how the flood inundation maps are different and imperfect leading to differences in estimated exposure and consequences. We think that the current analysis sufficiently proves this point.

The USGS includes a qualitative and quantitative uncertainty measure in the HWM data based upon the type of mark, the material the mark is upon, and the environment that creates the mark (Koenig et al., 2016). The quantitative height uncertainty attribute is present within the USGS HWM data and USGS associates that uncertainty qualitative quality of the data.

We have added the flooding source (coastal, riverine, or compound) to the summary of the HWM data in this section of the manuscript.

Line 198: intersecting rather than capturing (word choice)

Response: This change will be made in the manuscript.

Lines 199-200: from where was this assumption previously stated? Please state hypotheses in the introduction or methods sections.

Response: We have add this assumption to the introduction section.

Lines 210-226: predicting accurate WSE in HEC RAS is traditionally the model's aim, so this is also an important finding—and begs questions about why the model cannot be differenced into producing accurate inundation extents and depths

Response: The main inaccuracies associated with the HEC-RAS framework are associated with the modeler's choice of HEC-RAS geometry in Armand Bayou. Even with the limitations in accuracy created by the modeler's choice not to include distributed streamflow in the Armand Bayou geometry, the HEC-RAS framework is the most accurate in terms of WSE estimation. The HEC-RAS model did leave a larger amount of HWM points dry than the Fathom-US framework and most of these omissions where not due to the choice of model parameterization. However, even though the Fathom-US framework tended towards over-prediction, Fathom-US also left a portion of HWM data dry as well. This result appears to indicate how the use of multiple flood inundation mapping frameworks would be useful in communicating flood risk during a flood event, as they will all be imperfect. We have added to this discussion in the manuscript.

Line 218: what is MASL? Please make sure units and datums are presented consistently or explain the choices for one datum instead of another.

Response: Meters above sea level (MASL). We have converted this reference to meters (m) to avoid confusion on datums.

Line 225: again, where was this previously stated? There appears to be a set of assumptions that was not explained in the introduction or established as testable hypotheses

**Response: We have inserted this hypothesis into the Introduction section.**

Line 249: where was this expectation stated previously?

**Response: We have inserted this hypothesis into the Introduction section.**

Line 259: where was this expectation stated previously?

**Response: We have inserted this hypothesis into the Introduction section.**

Section 3.2: What is the source of flooding denoted by the HWM? Is there a distinction between riverine, coastal, or pluvial sources? Or are we to assume that the HWMs reflect compound flood conditions? Would one model perform better if only one source of flooding was evaluated by each model?

**Response: The USGS HWM data contain sourcing of coastal or riverine flooding. We've added an expanded discussion of the results to Section 3.2.**

Line 265: Why is Gesch et al 2014 cited for Fathom? Should this be a Wing citation? Or is the citation for some reference to NSSDA?

**Response: Gesch et al. (2014) is the source for the vertical accuracy estimate that precedes the citation.**

Lines 265-267: if HEC RAS is considered better accuracy, why does the derived inundation map not correlate with HWM intersections and/or depths? Are there other limitations to the model beyond parameterizations or lack of data?

**Response: The depths are a better estimated by the HEC-RAS framework (see Figure 4 and Table 2) but a lower proportion of HWM points are inundated by the HEC-RAS framework (Table 1). We've added this distinction to the manuscript.**

Line 269: Can Fathom run at 1-meter resolution to be commensurate with HEC RAS analysis? Comparing model resolutions is important to explain.

**Response: This is out of scope for this analysis. This manuscript aims to compare flood inundation map differences and if those differences result in quantifiable differences in exposure and consequences. We have noted here that spatial resolution of the DEM likely plays a role in the resulting flood inundation maps but further analysis is beyond the scope of this paper.**

Line 277: "the user must understand the assumptions made by the modeler." A better way to state this: "the user should understand the parameterizations made by the modeler." Choosing to omit or include certain parameterizations is the key message here and relates to the discussion in Section 3.3 at line 326— the user, and reader for that matter, needs the parameterizations. A user can make assumptions that one model is "better" than another based on same reported aspect of accuracy; however, the modeler's role is to express what is and what is not accounted for in an analysis—like, as lines 342-343 reflect, *not intending to represent flood inundation* because data is insufficient, erroneous, or non-existent. The modeler chose to not parameterize inundation for Armand Bayou in the HEC RAS analysis; therefore, the HEC RAS analysis should not be compared to the other models because it is incomplete, reflected in the statement at lines 343-344.

**Response: This change has been made in the manuscript.**

Lines 278-280: this appears to be explaining conditions very specific to HEC RAS, whereas the comment is directed to users of Event Maps—what other explanations of very specific modeling parameters or assumptions can be made more generally to apply to each of the models?

**Response: This is a question the author's have addressed in Section 3.4 of the manuscript. In general, the composition of the frameworks (e.g., Figure 2) should be presented in the metadata of each flood inundation map to assist with the vetting process. Also, within the metadata, a general descriptive narrative would be appropriate where the modeler can convey what they, using best professional judgement, think are appropriate specifics to convey to the user of the flood inundation map.**

Line 283-284: Why was AutoRoute chosen to model this explicitly compound flood event? Why not use a combination of other models to consider coastal vs riverine vs pluvial vs compound events?

**Response: AutoRoute was chosen as an example of the terrain filling flood inundation mapping frameworks, such as the Height Above Nearest Drainage (HAND) methodology. The intent of the manuscript was to gather a subset of various flood inundation mapping frameworks, estimate flood inundation extents with each, and assess if they are different and if those differences are consequential. Conventional wisdom would state that simplified riverine only flood inundation maps are inaccurate along the coast. However, they are still deployable and we intended to evaluate if the inaccuracies of the terrain filling, riverine only flood inundation maps would be consequential to our exposure and consequence estimates.**

Lines 300-315: This section is unclear, particularly lines 311-312 which appears to set out the overall differences in dollar/damage exposure: greater water depths should have greater expected losses, per the damage functions used in the study; however, little attention is given to water depths across the three modeled flood inundations. There is also no comparison of modeled depths to HWM depths. Line 313 suggests a bias in the AutoRoute model whereas greater depths may simply be a feature of the model or given its configuration for this analysis (e.g., it doesn't do coastal, so WSE will be higher given upstream/inland ground elevations and thus a potential for greater depths or depth errors from DEM).

**Response: Our apologies for the confusion!**

**Lines 298-305 summarize a general comparison of exposure and consequences from all three flood inundation mapping frameworks. These differences clearly demonstrate that the different spatial composition of the flood inundation maps leads to quantified differences in the exposure and consequences.**

**Lines 305-314 examine why AutoRoute inundates 6,279 structures while estimating $0.9 billion in damages while HEC-RAS inundates 19,281 structures while estimating $0.7 billion in damages (from Table 3). The relationship found is that when HEC-RAS and AutoRoute inundate the same buildings, AutoRoute estimates $0.3 Billion more in damages than HEC-RAS. The only explanation in this difference in damage is a higher depth, as go-consequences uses the same location and depth-damage function for these buildings. If we then look at structures where only HEC-RAS estimated damage, the sum total is $0.5 billion and the average water depth is 1.1 meters. Likewise, for only structures where AutoRoute estimates inundation and damage, the sum total is $0.3 billion and the average water depth is 3.8 meters. Thus, AutoRoute estimates more damage than HEC-RAS because of a tendency to estimate a higher water depth.**

We have clarified this in the manuscript.

Section 3.3: recommend not using the term "impact(s)" without discussing or getting into vulnerability assessment; recommend sticking to "exposure" to reduce confusion about the assessment

**Response: We have amended the manuscript based upon the reviewer's suggestions.**

Line 356: what is the "quantitative pattern" referenced here? Spatial pattern? Depths? Differences in elevations?

**Response: The proportion of insurance claims within each flood inundation map mirrors the proportion of HWM's within each flood inundation map. We have added this clarification to the manuscript.**

Line 360: Please clarify—is this the correct use of "deterministic" in this statement? It seems that the implication or operative term is single event, not single source. Merwade et al 2008 presents a method to display a single, deterministic (i.e. static) inundation map with possible spatial errors—that is, a flood inundation map that includes visualization of quantifiable uncertainties affecting the spatial extent of estimated flooding. Applying Merwade et al 2008 here infers that the maps produced by the 3 evaluated models each do not account for uncertainties that may include sources of flooding, different DEM resolutions and vertical errors, different roughness coefficients, etc. The follow-on reference to the national hurricane center interactions with stakeholders (NOAA 2013) does not refer to stakeholders favoring probabilistic storm surge maps and appears to conflate the approach offered in Merwade et al 2008 (that is, cartographic representation of uncertainty versus numerical or forecast uncertainties). The report states that stakeholders found the map colors and water depth classifications useful and easy to understand; however, the report details hazard-specific probabilistic maps (wind, storm location uncertainty, arrival timing of wind speeds—standard NHC advisory products) but not probabilistic storm surge inundation maps.

**Response: Excellent correction here! We intended to confer that one deterministic modeling chain leading to a flood inundation map will inherently possess imperfections/limitations and Table 1 is evidence of that and that multiple modeling chains that lead to multiple flood inundation maps may better confer risk. We have refined this section and associated references to improve the discussion of the point we want to make.**

Lines 369-374: Comparison of NFIP claims to NSI valuation is problematic and likely underestimates damages.

**Response: We have removed this comparison, given the problematic nature of comparing these datasets.**

Section 3.3 offers an interesting solution to a complex problem in producing and applying flood inundation maps in emergency management situations. It would be interesting to delve further into the reasons that inundation mapping is not a primary function of the federal agencies partnered in IWRSS or that any one entity does not produce an authoritative map, like NHC does for hurricane storm surges. (Are the authors implying that the NHC storm surge map should also be refereed?) However, this seems somewhat beyond the scope or intent of this paper unfortunately—but one can't help but wonder what the reasons are for NWS or USGS or USACE not producing real-time, publicly-accessible inundation maps beyond technical limitations. Is there a statutory reason for not producing inundation maps in real-time? Budgetary or staffing shortages? Clearly these data and maps can be made, and many in near-real-time, so

is adjudication the right solution over, say, accounting for mapping uncertainties cartographically and explaining the use cases for the maps and data?

Response: Our use of the term Event Map appears to have confused the reviewer. We will change our references from Event Map to reduce confusion. Each agency in IWRSS has their own means of producing flood inundation maps for emergency situations and distributing those in real time (e.g., lines 376-381, along with our three examples). The issue has historically been that each agency produced their respective flood inundation map without fully coordinating with the other agencies. To further complicate things, there can also be non-IWRSS flood inundation maps created during a flood event. The Fathom-US maps are one example of a non-IWRSS flood inundation map and there are likely a number of others that are local and regional. With the number of flood inundation maps that may be available for a given flood event, a way of consolidating, adjudication, and promoting the appropriate flood map for a given location seems to be the most logical first step. That is what the integrated Flood Inundation Mapping (iFIM) effort intends to do (Mason et al., 2020). The idea being that if, for instance, a U. S. Geological Survey flood inundation map for a given time and location is most appropriate, that map will be promoted by all of IWRSS to emergency managers and the public. The result is the authoritative, consolidated Event Map.

**References**

Follum, M. L., Vera, R., Tavakoly, A. A., and Gutenson, J. L.: Improved accuracy and efficiency of flood inundation mapping of low-, medium-, and high-flow events using the AutoRoute model, Natural Hazards and Earth System Sciences, 20(2), https://doi.org/10.5194/nhess-20-625-2020, 2020.

Follum, M. L., Tavakoly, A. A., Niemann, J. D., and Snow, A. D.: AutoRAPID: A Model for Prompt Streamflow Estimation and Flood Inundation Mapping over Regional to Continental Extents, JAWRA Journal of the American Water Resources Association, 80523, 1–20, https://doi.org/10.1111/1752-1688.12476, 2017.

Koenig, T. A., Bruce, J. L., O'Connor, J., McGee, B. D., Holmes, R. R., Hollins, R., Forbes, B. T., Kohn, M. S., Schellekens, M. F., Martin, Z. W., and Peppler, M. C.: Identifying and Preserving High-Water Mark Data, Techniques and Methods 3–A24, 2016.

Mason, R., Gutenson, J., Sheeley, J., and Lehman, W.: What's New (And What Does it Mean) – Technology Edition, St. Louis, MO, 25-28 February 2020 Interagency Flood Risk Management Program Workshop, 2020.

Shao, W., Xian, S., Lin, N., Kunreuther, H., Jackson, N., and Goidel, K.: Understanding the effects of past flood events and perceived and estimated flood risks on individuals' voluntary flood insurance purchase behaviour, Water Research, 108, 391–400, https://doi.org/10.1016/j.watres.2016.11.021, 2017.

**RC2**

Summary

In this study, the authors compare three modeling frameworks for mapping inundation extent and flood depth in Clear Creek watershed, a tributary of Galveston Bay in Texas, U.S. They evaluate the performance of AutoRoute, HEC-RAS, and Fathom-US frameworks against in-situ USGS high-water marks of Hurricane Harvey, a well-studied compound flood event in 2017. Also, the authors estimate flood exposure, consequences, and damages to buildings using available data from FEMA. It is shown that both HEC-RAS and Fathom-US outperform AutoRoute due to inherent limitations of the latter framework to simulate flooding in low-lying areas. Although Fathom-US and HEC-RAS achieve high location accuracies and low error and bias, they present some discrepancies regarding the evaluation metrics. The authors suggest an ensemble, multimodel probabilistic methodology to leverage these frameworks and provide more accurate flood maps as discussed in similar studies.

Major comments

This study presents an inter-model comparison with a practical application in terms of flood exposure and damage assessments, but it does not provide essential information for doing so. In contrast to Fathom-US (Wing et al., 2017, 2019), there is no evidence of model calibration and validation of both AutoRoute and HEC-RAS models for the study area. If the goal is to evaluate model's performance, then input data, forcing, mesh extent, and grid resolution should be identical among the frameworks. This compromises not only the validity of the results, but the analyses presented throughout the manuscript. I suggest the authors to consult or follow other studies that provide guidelines for model comparison (Shustikova et al., 2019; Muñoz et al., 2021; Afshari et al., 2018; Liu et al., 2018).

The authors investigate the performance of the frameworks knowing beforehand that AutoRoute is not suitable in coastal areas (Line 63 in the Introduction). This rises concern about the usefulness of a low-skill model in this study. If the authors want to consider steady-state models like AutoRoute (or HAND) in the model comparison, I suggest to follow the approach of Jafarzadegan et al., (2022) to enhance model simulations via hydrogeomorphic classifiers.

Response: We are thankful for the reviewer's comments, critiques, and suggestions. Unfortunately, the reviewer has misinterpreted the goals of this study. We intend for the evaluation of each flood inundation mapping framework to be used as a tool for discerning if and how the flood inundation maps differ in their spatial composition and if those differences lead to different estimates of exposure and consequences for a case study event (Hurricane Harvey). We have deliberately chosen maps with differing DEM resolution, streamflow forcing, and numerical schemes to simulate what would occur in reality during a Harvey-like flood event. Any three of these flood inundation mapping frameworks we chose for the study could be deployed for flood inundation mapping during a flood event. Understanding that each flood inundation map is different and that an emergency manager or the public could make different conclusions based upon those differences is critical for real-time flood inundation map coordination. The authors think that our manuscript offers evidence that the different quantification and spatial patterns of exposure and consequences produced using each flood inundation map from our study could lead a user, such as an emergency manager or member of the public, to draw different conclusions about the flood events impacts. The differences we observe justify the need to centrally vet and adjudicate flood inundation maps and promote an official map that should be used for a given time and location, using groups such as the integrate Flood Inundation Map (iFIM) team (Mason et al., 2020). The study also offers further evidence that use of multiple flood inundation maps offers utility over an individual map.

**Because of the reviewers misinterpretation of the focus of this study, we have revised the manuscript to better focus on the primary goals of the study. This includes a revision of the title and general wording of the manuscript.**

Minor comments

L16: 'Event maps' is too generic for referring to flood inundation maps. 'Event maps' are also used to describe the modeling framework making the manuscript difficult to follow in some sections.

**Response: We have revised the manuscript to refer to the flood inundation maps as FIMs. The companion reviewer also had some trouble with our reference of Event Maps.**

L20: Are you talking about modeling frameworks or flood inundation maps? How can event maps be physically different?

**Response: We are talking about flood inundation mapping frameworks. We have made this correction.**

**Apologies for any confusion, physical differences was a reference to the different spatial compositions of each flood inundation map. We have updated the manuscript to remove references to the maps physical differences.**

L26: Do you mean flood emergency response instead of flood fights?

**Response: Yes, the authors have made this change in the manuscript.**

L28: We find that the modeling frameworks are much different physically…

**Response: Apologies for any confusion, physical differences was a reference to their different spatial compositions. We have updated the manuscript to remove references to the maps physical differences and replace the reference with a more appropriate terminology.**

L43: Typo. Event Maps help emergency managers…10

**Response: We have addressed this error in the revised manuscript.**

L63: HAND can be adapted to simulate coastal flooding in low-lying areas. See Jafarzadegan et al., (2022).

**Response: We have added the Jafarzadegan et al., (2022) reference to the manuscript and distinguish between the traditional HAND methodology and the newer, revised versions that intend to improve flood inundation maps in low-lying, coastal regions.**

L88: I can anticipate that you will find substantial differences based on the DEM resolution and forcing data you have chosen for each framework.

**Response: We agree and this is the intended investigation of this manuscript. During Hurricane Harvey, any of these three flood inundation mapping frameworks could have been deployed to assist emergency management and response. The fact that each map is composed using different DEM resolution and forcing is something that will inevitably occur in real world scenarios. We intend for this manuscript to demonstrate that the different compositions of each flood inundation mapping framework (e.g., DEM resolution and forcing) can lead to resulting differences in the spatial composition of each flood inundation map estimate. These differences in each flood inundation map lead to different exposure and consequence estimates providing evidence that the**

**differences in each flood inundation map are substantive and that a central vetting and adjudication process for the flood inundation maps (e.g., the integrated Flood Inundation Mapping (iFIM) effort (Mason et al., 2020) is necessary for flood events.**

L98: Details of the hydrologic and hydraulic modeling are missing. For example, what is the grid size for the 2D component?

**Response: We have revised the manuscript to include any missing and necessary details of the hydrologic and hydraulic modeling.**

L120: Diffusive wave is a simplified version of the shallow water equations. Given the nonlinearities and complexities arising in compound coastal flooding, the complete set of equations (SWL) available in HEC-RAS should be used. This might lead to a better accuracy of the HEC-RAS in terms of inundation extent and flood depth.

**Response: An on-going Regional Flood Study effort, led by the Texas General Land Office, is evaluating how HEC-RAS model accuracy changes due to the usage of the Diffusion Wave equations and the original Shallow Water equations (SWE-ELM, which stands for Shallow Water Equations, Eulerian-Lagrangian Method). The preliminary findings of this analysis shows the differences between these two equation usages in HEC-RAS model prediction accuracy on inundation extent and depth are negligible in the upstream of the watershed, whereas minor differences exist, especially near the model downstream locations. Because this discussion is out-of-scope for this manuscript, given that the focus of the study does not focus on model evaluation, we have not added this discussion to the manuscript.**

L123: What are those mysterious downstream boundary conditions? Figure 1 should include the location of those boundaries for the three modeling frameworks.

**Response: The authors disagree with adding this detail to Figure 1. Inclusion of multiple boundary condition locations within Figure 1 will only cause Figure 1 to become illegible and adds little to the discussion of the main topic of the manuscript.**

L129: Are roughness values calibrated afterwards? These initial 1D and 2D roughness values are event-specific and have to be tuned for future flood events.

**Response: The hydrologic and hydraulic components of the HEC-RAS framework were calibrated for Hurricane Harvey and the 2016 Tax Day floods (Nielsen and Schumacher, 2020). Roughness values in the 1D portion of the modeled come from standard values described in the MAAPNext process that were based on the Harris County Policy, Criteria, and Procedures Manual (PCPM). Those values are consistent with recognized and accepted engineering standards. The basis of the 2D roughness coefficients is a combination of values developed by the Houston-Galveston Area Council (HGAC) and early calibration/testing efforts by the contracted model developer (Freese and Nichols, Inc., 2021). We can add this description to the manuscript.**

L134: Would it not be better to consider the 1-m DEM and so avoid inaccuracies due to DEM resolution? Previously, you suggest using observed meteorological data to avoid limitations in forecast skill...

**Response: This would be true if the primary motivation for this study was to evaluate the accuracy of each flood inundation mapping framework. However, we intend for this manuscript to demonstrate that the different compositions of each flood inundation mapping framework (e.g.,**

DEM resolution) can lead to resulting differences in the spatial composition of each flood inundation map estimate. These differences in each flood inundation map lead to different exposure and consequence estimates providing evidence that the differences in each flood inundation map are substantive and that a central vetting and adjudication process for the flood inundation maps (e.g., the integrated Flood Inundation Mapping (iFIM) effort (Mason et al., 2020) is necessary for flood events.

L150: Evaluation of simulated time series is very informative but missing in this study (e.g., timing and magnitude of peak water level). I strongly suggest assessing model's performance based on time series of available USGS (#08077637) and NOAA stations.

Response: We do not agree that further emphasis on evaluation of times series is necessary. The evaluation of each flood inundation modeling framework in our study intends on evaluating each flood inundation mapping framework to the extent that we prove that each resulting flood inundation map is of a different spatial compositions and that none of the maps prefect represent reality. We believe that we have successfully proven that each flood inundation mapping framework will produce a different flood map with the current evaluation process. A time series evaluation would be impactful if this study was considering the effects of hazard communication in our consequences assessment. However, we have chosen to evaluate the impact of differences in the peak flood inundation mapping on consequences and exposure, not the impact of hazard communication. Thus, evaluation of timing is beyond the scope of this manuscript.

L183: What are the upper and lower bounds?

Response: We have removed this section from the manuscript.

L185. Typo in the diagram. "Create Kernel density maps".

Response: We have made the appropriate correction to Figure 3.

L197: Diffusion wave does not solve the full mass balance and momentum equations and therefore might have influenced flood inundation extent and depth. In addition, the 1D portion of the model cannot provide 2D flood maps and consequently miss nearby high water marks.

Response: An on-going Regional Flood Study effort, led by the Texas General Land Office, is evaluating how HEC-RAS model accuracy changes due to the usage of the Diffusion Wave equations and the original Shallow Water equations (SWE-ELM, which stands for Shallow Water Equations, Eulerian-Lagrangian Method). The preliminary findings of this analysis shows the differences between these two equation usages in HEC-RAS model prediction accuracy on inundation extent and depth are negligible in the upstream of the watershed, whereas minor differences exist, especially near the model downstream locations. We have made a note of this in Section 3.3 of the manuscript.

L200. I cannot find the calibration and validation process in this manuscript. The same holds for AutoRoute model.

Response: We have added the details of the HEC-RAS framework calibration and validation to the manuscript in Section 2.1. Details on the setup of the AutoRoute framework are also provided in Section 2.1.

L218: USGS high-water marks are referenced with respect to NAVD88. There may be uncertainties added in the NAVD88 to MSL conversion process. How did the authors conduct the datum conversion? What is the vertical datum of the DEMs?

Response: All vertical elevations are based upon NAVD88. To reduce confusion we now refer to high-water mark and water surface elevation measurements in terms of meters.

L226: … and the steady state assumption. Also, AutoRoute is only forced by streamflow ignoring the contribution of coastal water level (e.g., storm-tide) to compound flooding (Figure 2).

Response: Yes, we haves inserted the description of the state assumption in this portion of the manuscript.

L228: Figure 4. Text size is too small.

Response: We have revised the text size in Figure 4.

L412: I agree that ensemble modeling is the way to go for better compound flood assessments. Nevertheless, I consider Figure 8 unnecessary in this study as you are not actually following this approach for simulating compound flooding due to Hurricane Harvey. A descriptive text is enough for future work in this regard.

Response: We have removed Figure 8 from the manuscript.

References

Afshari, S., Tavakoly, A. A., Rajib, M. A., Zheng, X., Follum, M. L., Omranian, E., and Fekete, B. M.: Comparison of new generation low-complexity flood inundation mapping tools with a hydrodynamic model, Journal of Hydrology, 556, 539–556, https://doi.org/10.1016/j.jhydrol.2017.11.036, 2018.

Freese and Nichols, Inc.: Lower Clear Creek and Dickinson Bayou Flood Mitigation Plan Hydraulic Technical Memorandum Final Report: Appendix C, 2021.

Jafarzadegan, K., Muñoz, D. F., Moftakhari, H., Gutenson, J. L., Savant, G., and Moradkhani, H.: Real-time coastal flood hazard assessment using DEM-based hydrogeomorphic classifiers, 22, 1419–1435, https://doi.org/10.5194/nhess-22-1419-2022, 2022.

Liu, Z., Merwade, V., and Jafarzadegan, K.: Investigating the role of model structure and surface roughness in generating flood inundation extents using one- and two-dimensional hydraulic models, 0, e12347, https://doi.org/10.1111/jfr3.12347, 2018.

Mason, R., Gutenson, J., Sheeley, J., and Lehman, W.: What's New (And What Does it Mean) – Technology Edition, St. Louis, MO, 25-28 February 2020 Interagency Flood Risk Management Program Workshop, 2020.

Muñoz, D. F., Yin, D., Bakhtyar, R., Moftakhari, H., Xue, Z., Mandli, K., and Ferreira, C.: InterModel Comparison of Delft3D-FM and 2D HEC-RAS for Total Water Level Prediction in Coastal to Inland Transition Zones, n/a, https://doi.org/10.1111/1752-1688.12952, 2021. Shustikova, I., Domeneghetti, A.,

Neal, J. C., Bates, P., and Castellarin, A.: Comparing 2D capabilities of HEC-RAS and LISFLOOD-FP on complex topography, 64, 1769–1782, https://doi.org/10.1080/02626667.2019.1671982, 2019.

Nielsen, E. R., and Schumacher, R. S.: Dynamical mechanisms supporting extreme rainfall accumulations in the Houston "tax day" 2016 flood, Monthly Weather Review, 148(1), 83-109, https://doi.org/10.1175/MWR-D-19-0206.1, 2020.

Wing, O. E. J., Bates, P. D., Sampson, C. C., Smith, A. M., Johnson, K. A., and Erickson, T. A.: Validation of a 30 m resolution flood hazard model of the conterminous United States, Water Resources Research, 53, 7968– 7986, https://doi.org/10.1002/2017WR020917, 2017.

Wing, O. E. J., Sampson, C. C., Bates, P. D., Quinn, N., Smith, A. M., and Neal, J. C.: A flood inundation forecast of Hurricane Harvey using a continental-scale 2D hydrodynamic model, Journal of Hydrology X, 4, 100039, https://doi.org/10.1016/j.hydroa.2019.100039, 2019.